# Zero-Shot Learning with Joint Generative Adversarial Networks

**Minwan Zhang** [1] , **Xiaohua Wang** [1,2], **Yueting Shi** [1,3] , **Shiwei Ren** [1,2] and **Weijiang Wang** [1,2,*]

1   School of Integrated Circuits and Electronics, Beijing Institute of Technology, Beijing 100081, China
2   Beijing Institute of Technology Chongqing Center for Microelectronics and Microsystems, Chongqing 401332, China
3   Yangtze Delta Region Academy of Beijing Institute of Technology, Jiaxing 314019, China
*   Correspondence: wangweijiangbit@163.com

**Abstract:** Zero-shot learning (ZSL) is implemented by transferring knowledge from seen classes to unseen classes through embedding space or feature generation. However, the embedding-based method has a hubness problem, and the generation-based method may contain considerable bias. To solve these problems, a joint model with multiple generative adversarial networks (JG-ZSL) is proposed in this paper. Firstly, we combined the generation-based model and the embedding-based model to build a hybrid ZSL framework by mapping the real samples and the synthetic samples into the embedding space for classification, which alleviates the problem of data imbalance effectively. Secondly, based on the original generation-method model, a coupled GAN is introduced to generate semantic embeddings, which can generate semantic vectors for unseen classes in embedded space to alleviate the bias of mapping results. Finally, semantic-relevant self-adaptive margin center loss was used, which can explicitly encourage intra-class compactness and inter-class separability, and it can also guide coupled GAN to generate discriminative and representative semantic features. All the experiments on the four standard datasets (CUB, AWA1, AWA2, SUN) show that the proposed method is effective.

**Keywords:** zero-shot learning; generalized zero-shot learning; GANs; feature generation methods





## 1. Introduction

Supervised classification has achieved great success in the research, but in this kind of classification, each class needs enough labeling training, and the learned classifier cannot deal with unseen classes [1]. To solve the above problems, the methods of few/one-shot learning [2–4], open set recogniton [5], cumulative learning [6], class-incremental [7] and open world [8] have been put forward. However, in the above methods, if unseen classes with no available tag instance appear in the test stage, the classifier still cannot determine their class tag. Therefore, zero-shot learning (ZSL) is proposed [9]. With the help of auxiliary information that contains descriptions of seen and unseen classes and the knowledge learned from training sets that belong to seen classes, sufficient labeled instances are provided [10]. ZSL methods can generate predictions for instances that belong to unseen classes despite that the seen and unseen classes are disjointed [11]; that is, given labeled training instances belonging to the seen classes, zero-shot learning aims to learn a classifier which can classify testing instances belonging to the unseen classes. From this definition, we can see that the general idea of zero-shot learning is to transfer the knowledge contained in the training instances to the task of testing instance classification. The label spaces covered by the training and the testing instances are disjoint. Thus, zero-shot learning is a subfield of transfer learning. In transfer learning [12], knowledge contained in the source domain and source task is transferred to the target domain for learning the model in the target task [13].

Since its birth, ZSL has become a fast-developing field in machine learning and has a wide range of applications in computer vision, natural language processing, and ubiquitous

computing [13]. Previous works for ZSL learn a space embedding function to implement the classification. According to the choice of embedding space, embedding-based methods can be divided into three categories: semantic space embedding methods, visual space embedding methods, and common space embedding methods [14]. They directly estimate the conditional distribution or mapping between visual features and their corresponding attributes. Semantic space embedding methods map visual features to semantic space directly. DeViSE [15] is one of the most representative models; it learns a linear mapping between image and semantic space using an efficient ranking loss formulation, and it is evaluated on the large-scale ImageNet dataset. However, using the semantic space as the embedding space means that the visual feature vectors need to be projected into the semantic space, which will shrink the variance of the projected data points and thus aggravate the hubness problem [16,17]. To alleviate the hubness problem, Li et al. [18] proposed a novel deep neural network-based embedding model (DEM). Although DEM uses the output visual feature space of a CNN subnet as the embedding space, which can alleviate the hubness problem to a certain extent, the inconsistency between the manifold of visual features and semantic features leads to the semantic gap. To solve the above-mentioned problem, Min et al. proposed a domain-specific embedding network (DSEN) [19] model, which considers the problem of semantic consistency and prevents the semantic relationship from being destroyed in the embedded space. Although the embedding-based method has been used and developed for a long time and is a very competitive zero-shot image classification method, due to the extreme imbalance in the number of training samples between seen and unseen classes, most of the existing methods still have great limitations.

Recent works mainly focus on synthesizing image features with a generative model, and generation-based methods have become a hot research topic [20,21]. These methods fall into the data augmentation-based category. The basic assumption of approaches in this category is that the intra-class cross-sample relationship learned from seen classes can be applied to unseen classes. Once the cross-sample relationship is modeled and learned from seen classes, it can be applied on the unlabeled samples of unseen class to hallucinated new samples, and unsupervised learning is transformed into supervised learning using synthesized new samples [22].

Depending on the different generation models, the existing generation-based methods mainly include GAN-based methods, VAE-based methods, and normalizing flow-based methods [23–25]. The normalizing flow-based methods build complex distributions by mapping a simple distribution through invertible functions, and they allow exact likelihood calculation while being efficiently parallelizable, but they have not been widely studied due to the particularity of the architecture [25]. Most of the VAE-based methods are unidirectional alignment. This method captures the low-dimensional potential features of visual features and then realizes unidirectional alignment between the generated pseudo-visual features and semantic attributes through decoding and reconstruction of the formula. SE-GZSL [26] adopts the VAE-based structure, and the generation model is composed of the probabilistic encoder and conditional decoder. At the same time, the feedback drive mechanism is introduced, which can improve the reliability of the generator. Although VAE is capable of generating pseudo-visual features stably to effectively avoid pattern collapse, the semantic information contained in the generated pseudo-visual features is very limited. In order to overcome the above problems, the GAN-based methods are proposed; this method can generate high-quality pseudo-visual features after the model is trained. VERMA et al. [27] proposed a meta-learning model ZSML based on the class attribute condition setting. The generator module and discriminator module with a classifier were associated with the meta-learning agent, respectively, and the model could be trained only by inputting a few visible class samples. Xian et al. [28] use the generative adversarial network to make the classification based on semantic features and Gaussian noise to generate unseen visual features, transforming the zero-shot learning problem into a supervised classification problem. The result of generation-based methods is better than embedding-based methods, and it is also the mainstream method at present.

In the latest work in 2022, both embedding-based and generation-based methods have been further explored and updated. Xu et al. [29] propose a Visually Grounded Semantic Embeddings model (VGSE), which learns visual clusters from seen classes and automatically predicts the semantic embeddings for each category by building the relationship between seen and unseen classes given unsupervised external knowledge sources. In terms of generation-based methods, to generate high-quality and diverse image features, Yu et al. [12] proposes a new generative model that adds a semantic constraint module and introduces a Euclidean distance loss for constraining feature generation. Although the above methods can solve the problem of the existence of zero-shot learning, it also introduces a new problem: previous work on the generation-based methods only used one generative adversarial network to simulate the visual features of unseen classes and ignored the distribution of these generative features in the mapping space. This may make the semantic mapping point of the generated feature closer to the semantic prototype of the seen class in the semantic space, resulting in the final classification result still having a bias toward the seen class.

To obtain the best of both worlds and solve the new problem mentioned above, we first propose a hybrid model, which can implement both the space embedding-based method and the generation-based method. Second, we introduce a generation adversarial network to simulate the mapping point of unseen class features in the embedding space. Although the model with multiple GAN cascaded has been fully proven and used in supervised learning, it has not been applied to zero-shot learning. In this paper, a multilevel GAN stack structure is introduced for the first time in zero-shot learning to optimize the problem of data imbalance. Third, we propose a semantic-relevant self-adaptive margin center loss for the coupled GAN. This loss can encourage intra-class compactness and inter-class separability and realizes that the coupled GAN can better generate representative and differentiated semantic features. We evaluate our method on four benchmark datasets, and the experimental results show that our approach is competitive with other methods.

The contributions of this paper are summarized as follows:

- A hybrid model with joint generative adversarial networks (JG-ZSL) combining the embedding-based method and the generation-based method is proposed to improve model sensitivity and specificity.
- A GAN for generating semantic features is introduced to generate mapping points in embedding space, which can generate semantic vectors for unseen classes in semantic space to alleviate the bias of mapping results.
- Semantic-relevant self-adaptive margin center loss (SEMC-loss) is designed for the semantic generated GAN to ensure the generated mapping points in semantic embedding space are not biased to other categories and realize that the whole model can better distinguish between different classes.
- We evaluate our model on four benchmarks, and the experimental results show that our proposed method can achieve high accuracy.

## 2. Materials and Methods

### 2.1. Problem Definition

We have two disjoint sets of classes in both ZSL and GZSL: the seen class set $S = \{c_i^s | i = 1, \ldots, N_s\}$, where $c_i^s$ is a seen class which provides labeled instances for training, and unseen class set $U = \{c_i^u | i = 1, \ldots, N_u\}$ contains unlabeled instances for testing. Note that $S \cap U = \varnothing$. These instances have different visual features, but for instances from the same class, their labels and semantic descriptions are the same. Denote the visual feature as $x$, class label as $y$, and semantic description, which is the attribute in this article as $a$. Then, each class can be represented as a set $C_i = \{(x_i^j, y_i^j, a_i) | i = 1, \ldots, N_S + N_u; j = 1, \ldots, n\}$, $n$ is the number of instances the class contains; we can infer the semantic descriptor $a$ for an instance $x$ from its class label $y$.

ZSL aims to learn a classifier that can categorize the testing instances $x_u$ belonging to the unseen classes $U$, $f_{zsl} : x_u \rightarrow U$. Under the more challenging generalized zero-shot

learning (GZSL) setting, the testing instances $x$ come from both seen class $S$ and unseen classes $U$ because people are also concerned with the ability to classify instances on seen and unseen classes. GZSL aims to learns a classifier $f_{gzsl} : x \rightarrow S \cap U$.

Zero-shot learning is divided into three learning settings by Wang [13] et al. according to whether unlabeled testing instances and the class description information of the unseen class are used in model learning, as shown in Table 1. In this paper, unlabeled testing instances are not used when training the generators, but the classifier is trained using average visual features of unlabeled testing instances and synthetic features that the generator generates based on the attribute descriptions of the unseen classes. According to Wang's definition, our method belongs to the Class-Transductive Instance-Transductive (CTIT) Setting.

**Table 1.** Zero-Shot learning setting.

| | | |
|---|---|---|
| whether unlabeled testing instances are used | yes | Instance-Transductive |
| | no | Instance-Inductive |
| whether description information of unseen class are used | yes | Class-Transductive |
| | no | Class-Inductive |

### 2.2. Hybrid Framework Introduction

The proposed joint GAN cascaded for ZSL (JG-ZSL) is illustrated in Figure 1. Specifically, the network consists of an embedded network that maps visual features to semantic space, a GAN that generates visual features based on attributes, and a GAN network that generates semantic space mapping points based on visual features.

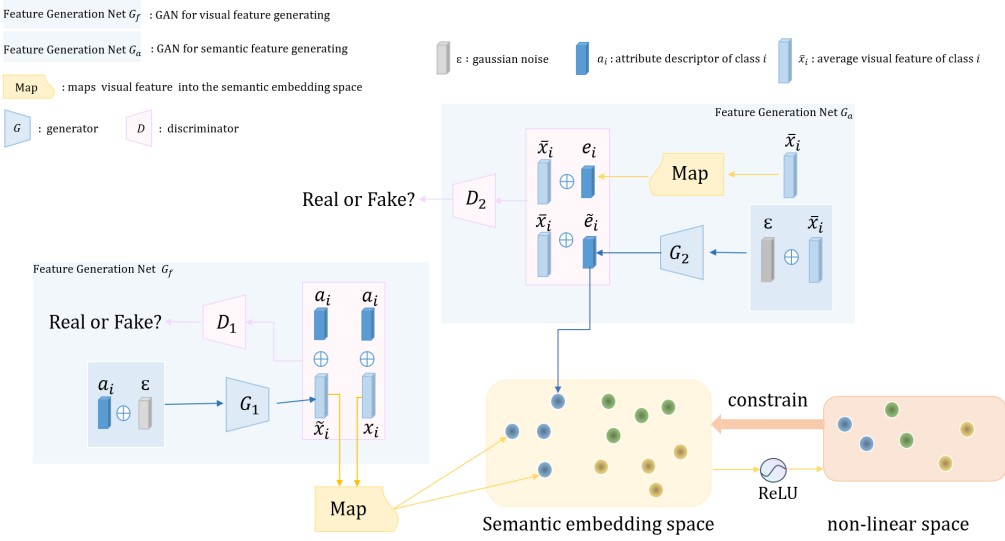

**Figure 1.** Scheme of the proposed joint GANs (JG-ZSL) network.

### 2.2.1. Mapping Net

Human beings can summarize the attributes of the observed objects according to the visual features seen by the naked eye and deduce the categories of the observed objects according to the attributes. For example, if a child learns from watching a horse, a panda, and a tiger that they are "horse-like", "black-white", and "striped", he or she can easily distinguish a zebra from a variety of animals after being told that a zebra is a horse with black and white stripes [30]. This ability to recognize objects without any visual samples, only prior knowledge, is zero-shot learning. It is very necessary to ensure machines have the zero-shot learning ability: first, in real life, the object categories to be recognized usually

follow the long-tail distribution, some of which have rich training samples, while others have few or no available training samples. Zero-shot learning can not only get rid of the dependence on a large number of manual labeling samples but also have high commercial value in some applications lacking labeling samples. Hence, in order to enable machines to have this capability, ref. [9] introduced a manually defined attribute layer for the first time. Through this attribute layer, the classifier based on low-dimensional image features is transformed into a classifier based on high-dimensional semantic features (attribute layer) so that the trained classifier has broader classification ability and the ability to break through category boundaries. For example, in an animal identification problem in an image, attributes can be a body color (for example, "gray", "brown", and "yellow") or habitat (for example, "coastal", "desert", and "forest"). These attributes are then used to construct semantic spaces.

Semantic embedding (SE) in conventional ZSL aims to learn an embedding function $E$ that maps a visual feature $x$ into the semantic embedding space denoted as $h = E(x)$. The embedding function $E$ is usually a linear transformation consisting of two liner layers, whose input dimension is set to the dimension of the visual feature and output dimension is set to the dimension of the semantic feature. At the same time, $h = E(x)$ is also called linear semantic space because it is composed of fully connected layers. These commonly used semantic embedding methods rely on a structured loss function proposed in [15]. According to the dot product similarity in the embedding space, the structured loss requires that the embedding of $x$ is closer to the semantic embedding $a$ of its ground-truth class than the other class embeddings. Specifically, the structured loss formula is as follows:

$$\mathcal{L}_{SE}(E) = \mathbb{E}_{p(x,a)}[\max(0, \Delta - a^T E(x) + (a')^T E(x))] \tag{1}$$

where $p(x, a)$ is the empirical distribution of the training samples of seen classes, $a'$ is a random selection semantic descriptor of the other categories except $a$, and $\Delta > 0$ and is a margin parameter to make $E$ more robust.

On the basis of the traditional embedding function, Chen et al. [31] found that adding a non-linear projection head $H$ in embedding space as $z = H(h)$ can better constrain the original linear embedding space $h = E(x)$, because they showed experimentally that more information can be formed and maintained in $h$ through this non-linear projection. In the same way that $h = E(x)$ is called linear space because $E$ is composed of fully connected layers, we called $z = H(h)$ a non-linear space because the projection $H$ actually is a ReLU non-linearity. We follow Chen's strategy in our model; the difference is, Chen set $H$ and $E$ with the same output dimensionality (e.g., 2048-d), while we change the output dimension of $E$ to the dimension of the semantic descriptor of the dataset (e.g., for dataset CUB, 312-d); then, the linear space can be limited to the semantic embedding space.

For the non-linear space $z = H(h)$, we follow the strategy in [32] to perform the $(K + 1)$-way classification on $z_i$ to learn the embedding $h_i$, where $K$ is the number of negative examples $h_i^-$, which refers to the samples whose class label is different from the class label of $h_i$, while the only one positive example is $h_i^+$. Concretely, the cross-entropy loss of this $(K + 1)$-way classification problem is calculated as follows:

$$\mathcal{L}_{SE}(H) = -\log \frac{\exp(z_i^T z^+ / \tau_e)}{\exp(z_i^T z^+ / \tau_e) + \sum_{k=1}^{K} \exp(z_i^T z_k^- / \tau_e)} \tag{2}$$

where $\tau_e$ is a constant called the temperature parameter, which is manually set to adjust the degree of attention paid to negative samples. The smaller the temperature parameter is, the more attention is paid to separating this sample from other samples that are most similar.

### 2.2.2. Feature Generation Nets

The main disadvantage of embedding-based methods is that they suffer from the bias problem. This means that since the projection function is learned using only seen

classes during training, it will be biased to predict with seen class labels as output; this bias problem is caused by a serious data imbalance between seen and unseen class data.

In supervised learning, the problem of data imbalance refers to the huge difference in the number of samples in each category of the dataset. Take the binary classification problem as an example: assuming that the number of samples of the positive class is much larger than that of negative class, in this case, the data are called unbalanced data. In zero-shot learning, this problem is even more extreme; that is, part of the class samples as unseen classes are completely missing and cannot participate in the model training process. Therefore, in supervised learning, the method of repeatedly sampling categories with fewer samples (over-sampling) or reducing sampling for categories with more samples (under-sampling) to achieve data balance is not applicable to zero-shot learning. After all, no samples can be collected from unseen classes. Therefore, unseen class data generation has become a hot research topic, which can generate pseudo-samples for unseen classes, so that both seen and unseen classes have training samples and transform unsupervised learning into supervised learning. Generative Adversarial Networks [23] are particularly appealing as they allow generating realistic and sharp images conditioned, for instance, on object categories. Previous work on generation-based methods learn a generation network to produce the unseen sample. However, in previous work on generation-based methods, the synthesized instances are usually assumed to follow some distributions (usually Gaussian distribution) [13], which also leads to a large deviation between the generated sample and the real sample, and it cannot truly represent the real data situation of the unseen class. The idea of stacking multilevel generation networks has been proven to be effective in improving the quality of generation quality, but it has not been used in the ZSL field. In this paper, two conditional GANs ($G_f$ and $G_a$) are stacked to solve the problem of data imbalance from different aspects.

$G_f$, *the GAN for generating visual feature* : The network based on traditional GAN takes random noise as the prior information input, and the inherent randomness of the deep neural network makes the quality of the image generated by it unstable. To solve this problem, conditional GAN is proposed. By adding conditional information to the network model, it guides the network model to generate pseudo-samples matching the conditions. We extend the GAN to a conditional GAN by integrating the class embedding to both the generator $G_1$ and the discriminator $D_1$. Given the training data of seen classes, $G_1$ takes random Gaussian noise $\varepsilon$ and semantic embedding $a_y$ as its inputs and outputs a CNN image feature $\tilde{x}$ of class $y$. Once the generator $G_1$ learns to generate CNN features of seen class images, i.e., $x$, conditioned on the seen class embedding $a_s$, it can also generate $\tilde{x}$ of any unseen class via its class embedding $a_u$. The objective function can be expressed as:

$$\min_{G_1} \max_{D_1} V(D_1, G_1) = E[\log D_1(x, a_y)] + E[(1 - \log D_1(\tilde{x}, a_y))] \tag{3}$$

However, the adversarial nature of GANs makes them notoriously difficult to train, and the Jenson–Shannon divergence optimized by the original GAN leads to instability issues. To cure the unstable training issues of GANs, Wasserstein-GAN (WGAN) [33] is proposed, which optimizes an efficient approximation of the Wasserstein distance [25]. While WGAN attains better theoretical properties than the original GAN, it still suffers from vanishing and exploding gradient problems due to weight clipping to enforce the 1-Lipschitz constraint on the discriminator. So, we use the improved variant of WGAN, that is, WGAN-GP [34], which can enforce the Lipschitz constraint through gradient penalty. We extend the original WGAN-GP to a conditional WGAN-GP by integrating the class embedding $a_y$ to both the generator and the discriminator.

The loss is,

$$\mathcal{L}_{WGAN_{feature}} = E[D_1(x, a_y)] - E[D_1(\tilde{x}, a_y)] - \lambda E[(||\nabla_{\hat{x}} D_1(\tilde{x}, a_y)||_2 - 1)^2] \tag{4}$$

where $\tilde{x} = G_1(a_y, \varepsilon)$, $\hat{x} = \alpha x + (1 - \alpha)\tilde{x}$ with $\alpha \in U(0, 1)$, and $\lambda$ is the penalty coefficient. In contrast to the traditional GAN, the discriminative network here eliminates the sigmoid layer and outputs a real value. Instead of optimizing the log-likelihood in Equation (3), the first two terms in Equation (4) approximate the Wasserstein distance, and the third term is the gradient penalty which enforces the gradient of $D_1$ to have a unit norm along the straight line between pairs of real and generated points.

$G_a$, *the GAN for generating semantic embedding*: The embedding-based method obtains labeled instances of unseen classes by mapping instances in feature space and attribute prototypes in semantic space into the same space. Feature space contains labeled training instances of seen classes, and semantic space contains attribute prototypes of seen and unseen classes. Both spaces are real number spaces in which instance and attribute prototypes are vectors, respectively. By projecting the instance vectors from these two spaces into a common space, we can obtain labeled instances of unseen classes and classify them in the mapping space. However, in the embedding-based method, for every unseen class $c_u$, it has no labeled instance in the feature space; thus, its attribute prototype $a_u$ in semantic space is the only labeled instance belonging to the unseen class. That is, only one labeled instance is available for each unseen class. Therefore, since there are few label instances of unseen classes, the feature generation methods are proposed to solve the problem of data imbalance by generating visual features for unseen classes in feature space. However, in semantic space, labeled instances of the unseen class are still scarce. Especially in the GZSL setting, the mapping results are still biased toward the seen class. Therefore, appropriately adding semantic vectors of unseen classes in semantic space can alleviate the bias of mapping results.

Active learning is similar to zero-shot learning to some extent. Both of them are designed to reduce the dependence on large-scale labeling data and are targeted at scenarios where labeled data are rare or the "cost" of labeling is high. The difference is that zero-shot learning aims to realize knowledge transfer in the absence of labeled samples, while active learning aims to maximize model performance by actively selecting the most valuable samples for labeling. Therefore, some techniques in active learning can enlighten us. In active learning, Parvaneh et al. proposed the feature mixing method: compute the average visual representation $\bar{x}$ of the labeled samples per class and call it an anchor. The anchors for all classes form the anchor set $\bar{x}$ and serve as representatives of the labeled instances [17]. Inspired by their method, we take the average visual feature as a representation of one class and generate the semantic embedding $\tilde{e}$ of how this class might be mapped, as shown in Figure 2. The generated semantic embedding $\tilde{e}$ should have the following two characteristics. First, by generating the different semantic embeddings that may be mapped from the same class, we extend the original unique semantic discriptor of each category in the semantic space into a semantic discriptor a set $S_i = \{a_i, \tilde{e}_1, \ldots, \tilde{e}_n\}$, wherea$_i$ is the real semantic discritptor of category $i$ provided by the dataset, while $\tilde{e}_1$ to $\tilde{e}_n$ are the synthetic pseudo-semantic-discriptors just like extending the unique evaluation criteria to establish a qualifying interval. Second, generated semantic embeddings should be representative and authentic, which are similar to the semantic embeddings of the real existence mapped by the visual feature, and they can truly simulate the possible mapping situation without deviating from reality. We formulate our assumption for the pseudo-semantic embedding generation method as follows:

$$\tilde{e}_i = G_2(\varepsilon, \bar{x}_i) \tag{5}$$

$$\bar{x}_i = \frac{1}{n} \sum_{j=1}^{n} x_i^j \tag{6}$$

where $n$ is the number of visual features instances the class $i$ contains, and $x_i$ is the set of visual features contained in class $i$.

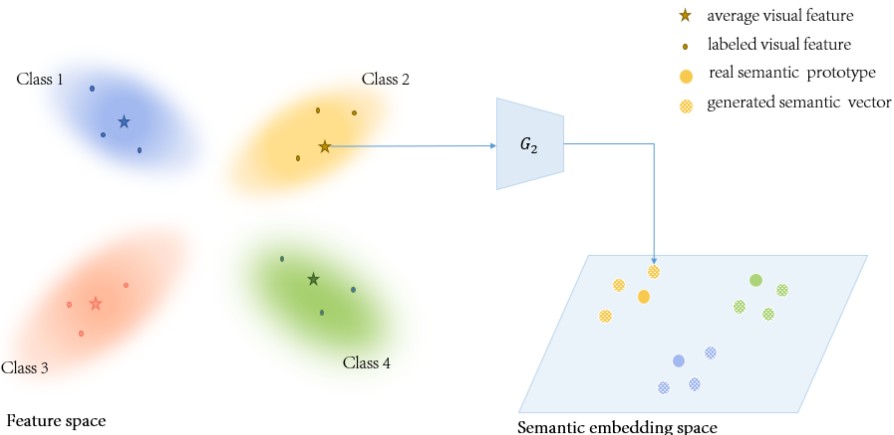

**Figure 2.** Scheme of pseudo-semantic embedding.

We still select the condition WGAN by integrating the visual feature average $\bar{x}$ to both the generator and the discriminator. The loss is,

$$\mathcal{L}_{WGAN_{att}} = E[D_2(e_i, \bar{x}_i)] - E[D_2(\tilde{e}_i, \bar{x}_i)] - \lambda E[(||\nabla_{\hat{e}_i} D_2(\tilde{e}_i, \bar{x}_i)||_2 - 1)^2] \qquad (7)$$

where $e_i$ is, corresponding to the synthetic semantic embedding $\tilde{e}_i$, the real semantic embedding obtained by inputting the average visual features $x_i$ of category $i$ into the mapping net.

*2.3. Loss Design*

2.3.1. Semantic-Relevant Self-Adaptive Margin Center Loss

To encourage $G_{att}$ to generate more representative semantic embedding for an unseen class, we used the idea of building a distance metric in metric learning. Metric learning aims to learn such a distance metric for a type of input data that conforms to semantic distance measures between the data instances [35]; this point has been explored and applied in both few-shot learning [35] and zero-shot learning [36,37]. Inspired by previous work, we propose the semantic-relevant self-adaptive margin center loss ($SEMC - loss$, $\mathcal{L}_{SEMC}$) to constraint $G_{att}$. By narrowing the distance between the generated semantic vector and the real semantic vector in the semantic space, intra-class compactness and inter-class separability are encouraged. It has the advantages of the center loss [38] and triplet loss [39] as well as learning intra-class compactness and inter-class separability. $\mathcal{L}_{SEMC}$ is formulated as:

$$\mathcal{L}_{SEMC} = \max(0, \Delta + \gamma||\tilde{e}_i - a_i||_2^2 - (1 - \gamma)||\tilde{e}_i - a_{i'}||_2^2) \qquad (8)$$

where $a_i$ is the $i$ th (the label of seen visual feature $x$) class center of semantic embedding, $a_{i'}$ is the $i'$th (a randomly selected class label other than $i$) class center, $\Delta$ represents the margin that $i$ controls the distance between intra- and inter-class pairs, $\tilde{e}_i$ is the synthesized semantic embedding of the $i$th class generated by $G_{att}$ and $\gamma \in [0,1]$ is used for balancing the inter-class separability and intra-class compactness, which are adaptable to various datasets. The sensitivity of intra-class compactness and inter-class separability to different datasets (coarse-grained datasets and fine-grained datasets) can be satisfied by using balance factors to balance intra-class separability and intra-class compactibility adaptively. We use a large $\gamma$ for fine-grained datasets (e.g., CUB [40], SUN [41]) and a small $\gamma$ for coarse-grained datasets (e.g., AWA1 [9], AWA2 [42]). For fine-grained datasets, we can more easily distinguish them by encouraging intra-class compactness, and for the coarse-grained datasets, we can effectively separate them by enlarging the inter-class separability.

2.3.2. Total Loss

In our hybrid framework, we map both the real features and the synthetic features into the semantic embedding space, where we perform the final GZSL classification. Notably, we formulate $\mathcal{L}_{SE}(E)$ only using the semantic descriptors of seen classes. Therefore, Equation (1) should be extended to:

$$\begin{aligned}
\mathcal{L}_{SE}(E) = \ &\mathbb{E}_{p(x,a)}[\max(0, \Delta - a^T E(x) + (a')^T E(x))] \\
&+ \mathbb{E}_{p_{G_f}(\tilde{x},a)}[\max(0, \Delta - a^T E(G_f(a,\varepsilon)) + (a')^T E(G_f(a,\varepsilon)))]
\end{aligned} \quad (9)$$

where $p(x,a)$ is the empirical distribution of the real training samples of seen classes, and $p_{G_f}(\tilde{x},a) = p_{G_f}(\tilde{x}|a)p(a)$ is the joint distribution of a synthetic feature and its corresponding semantic descriptor.

The total loss of mapping net takes the form of:

$$\mathcal{L}(G_1, E, H) = \mathcal{L}_{SE}(E) + \mathcal{L}_{SE}(H) \quad (10)$$

Thus, the total loss of our final hybrid framework is formulated as:

$$\mathcal{L}_{total} = \mathcal{L}(G_1, E, H) + \mathcal{L}_{WGAN_{feature}} + \mathcal{L}_{WGAN_{att}} + \mathcal{L}_{SEMC} \quad (11)$$

*2.4. Classification*

First, given the average visual representation $\bar{x}$ of the unlabeled samples per unseen class, we generate semantic features for each unseen class $c_u$ by the feature generator network $G_2$, which uses the average visual representation $\bar{x}$ and Gaussian noise as input and output synthetic features: $\tilde{e}_u = G_2(\bar{x}, \varepsilon)$. Second, to keep the inputs of the classifer in the same model, we use the $G_1$ to generate visual features for each pseudo-semantic embedding, that is, $G_1$ uses real semantic features and generated semantic features, respectively, to synthesize visual features, which are denoted as $\tilde{x} = G_1(a_u, \varepsilon)$ and $\tilde{x}' = G_1(\tilde{e}_u, \varepsilon)$. Then, we can obtain a synthetic training feature set $U_{tr} = \{\tilde{x} \cup \tilde{x}'\}$.

In the end, we map the synthetic training feature set $U_{tr}$ and the given training features of seen classes in $S_{tr}$ into the same embedding space $h_i = E(x_i)$ and utilize the real seen samples and the synthetic unseen samples in the embedding space to train a softmax model as the final classifier. The whole process is shown in Figure 3.

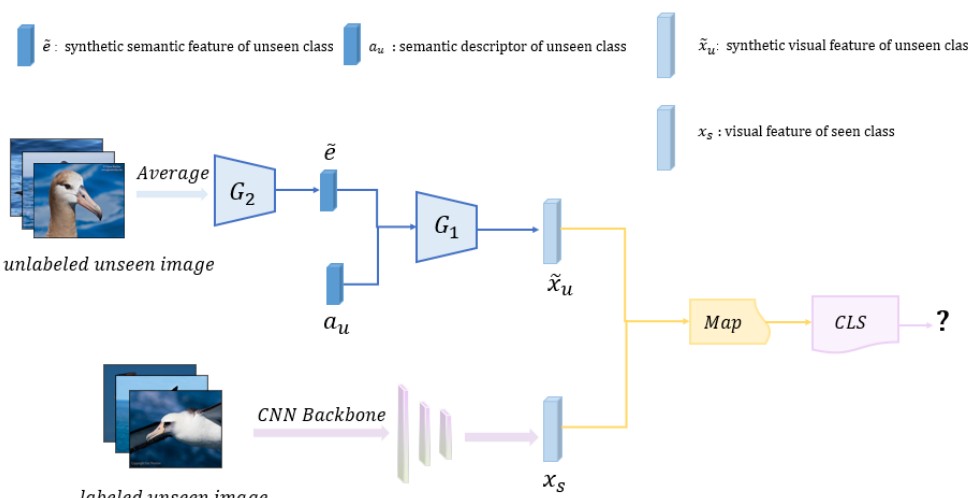

**Figure 3.** Scheme of classification.

## 3. Experimental Results

### 3.1. Datasets

We evaluate our method on four benchmark datasets for ZSL: Animals with Attributes 1 and 2 (AWA1 [9] and AWA2 [42]), Caltech-UCSD Birds-200-2011 (CUB) [40], and SUN Attribute (SUN) [41]. An example of the contents of each dataset is shown in Figure 4, all datasets and their statistics are summarized in Table 2 .

AwA1 is a coarse-grained image dataset, containing 30,475 animal pictures in 50 categories, 40 of which are the seen class and 10 of which are the unseen class, and 85-dimensional class level attribute vectors are used. AWA2 is a fixed version of AWA1; they have the same category, category division way, and class-level attribute dimension, except that 37,322 coarse-grained animal pictures are used, and they do not overlap with AwA1 image instances.

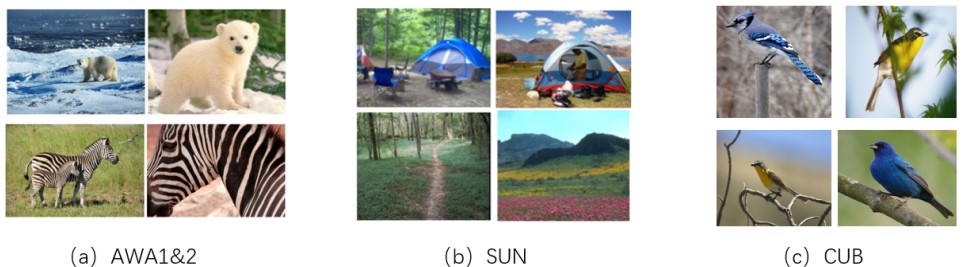

|(a) AWA1&2 | (b) SUN | (c) CUB |

**Figure 4.** Scheme of classification.

CUB is a fine-grained image dataset, including 11,788 bird pictures in 200 classes, of which 150 classes are in the seen class and 50 classes are in the unseen class. CUB also provides an instance-level attribute vector; however, only 312-dimensional class-level attribute vectors are used in this work. The class-level attribute descriptor space is shown in Figure 5.

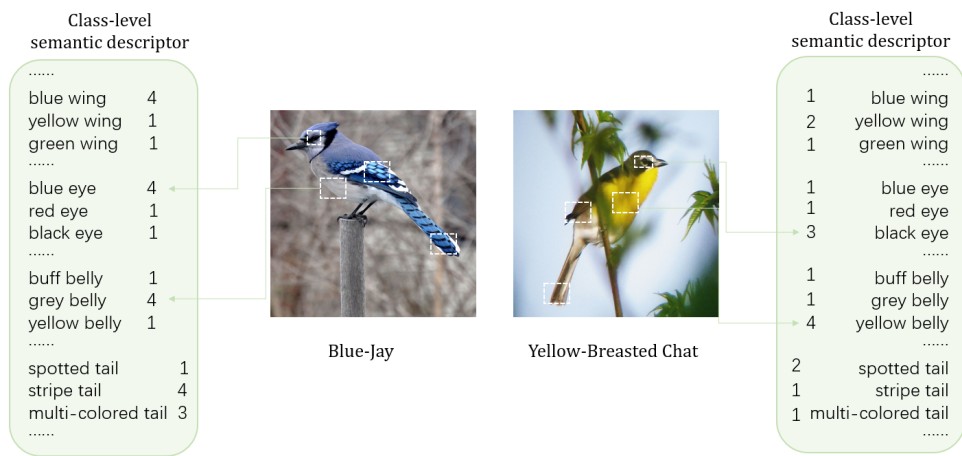

**Figure 5.** Example of attribute space for the CUB dataset.

**Table 2.** Statistics of the four benchmark datasets used in our experiments.

| Dataset | Seen/Unseen Class | Attribute | Train Seen | Test Seen | Test Unseen | Total Instance |
|---------|-------------------|-----------|------------|-----------|-------------|----------------|
| AWA1 | 40, 10 | 85 | 19,823 | 4958 | 5685 | 30,475 |
| AWA2 | 40, 10 | 85 | 23,527 | 5882 | 7913 | 37,322 |
| CUB | 150, 50 | 312 | 7057 | 1764 | 2967 | 11,788 |
| SUN | 645, 72 | 102 | 10,320 | 2580 | 1440 | 14,340 |

SUN is a scene dataset; this dataset is also a fine-grained one, which contains 14,340 pieces of 717 scenes. Here, 645 classes are used for training, and 72 classes are used for testing. Each class is annotated with a 102-dimensional attribute vector.

### 3.2. Implementation Details

We evaluated our method under the new split setting provided by [42], and more details on the settings can be found in [42]. We use strictly the 2048-dimensional feature of each image extracted from the pre-trained ResNet-101 provided by [42] similar to the others, and only the attribute vectors provided by each dataset are used.

We implement our method with PyTorch. We set the dimension of embedding $h$ to the class-level attribute vector, 85 for AWA1 & AWA2, 312 for CUB, and 102 for SUN. The dimension of the non-linear projection's output $z$ is set to 512. We set a random mini-batch size of 4096 for AWA1 and AWA2, 2048 for CUB, and 1024 for SUN. Our generator and discriminator both contain a 4096-unit hidden layer with LeakyReLU activation. The classification part contains one fully connected layer, which will be utilized in making predictions. The numbers of input and output units follow the dimension of attribute vectors and the number of classes provided by each dataset.

For the hyperparameter, we set the temperature parameter $\tau_e$ in Equation (3) according to [26]: $\tau_e = 0.1$ for AWA1, CUB and SUN, and $\tau_e = 10.0$ for AWA2. For the parameter in Equation (8), we use a large $\gamma = 0.8$ for fine-grained datasets (CUB and SUN) and a small $\gamma = 0.1$ for coarse-grained datasets (AWA1 and AWA2), referring to [43].

### 3.3. Experiments on Different Datasets

3.3.1. Performance of Different Methods under Comparison

Under the conventional ZSL scenario, we only evaluate the per-class Top-1 accuracy on unseen classes. The average per-class T1 accuracy is measured as follows, where $y$ represents the number of unseen classes and $c$ represents the serial number of each class:

$$acc_y = \frac{1}{||y||} \sum_{c=1}^{||y||} \frac{correct\ predictions\ in\ c}{samples\ in\ c} \tag{12}$$

To show the effectiveness of the proposed method, we compared the simulated results with six other algorithms, and all the results are cited directly from their published papers. To provide a fair comparison, we adopt the experiment settings provided by [42], i.e., the datasets and their splits, and all the algorithms we compared adopt the same experiment settings. Table 3 shows that our method achieved a high value for CUB and the second-best position for AWA1. On the AWA1 dataset, the MG-ZSL yields a Top-2 accuracy of 70.6%, while the best Top-1 accuracy is 73.5% (yielded by ZMSL). It is worth noting that the MG-ZSL yields Top-1 accuracy higher than 70% on the CUB datasets, which is 0.7% higher than the second-banked algorithms. These results show that the method presented in this paper has achieved remarkable results.

In general, the experimental results of this paper have considerable performance with the current best case and are significantly better than previous mapping-based methods, such as DeViSE and DEM. Moreover, it also surpasses SE-GZSL, which is one of the state-of-the-art generation-based methods for all datasets and is comparable to the ZSML. Thus, the MG-ZSL model is very competitive.

The conventional ZSL scenario has been criticized as a restrictive setup because it is based on a strong assumption that the instances used in the test stage only come from unseen classes, which is less realistic. Therefore, GZSL was proposed, which is more realistic in practice. In the GZSL setting, the instances for evaluation may come from seen and unseen classes, so we choose the harmonic mean as our main evaluation indicator instead of the arithmetic mean, because considerably high-class accuracy will significantly

affect the overall results with the latter. The harmonic mean can be computed by the following function:

$$H = \frac{2 \times acc_u \times acc_s}{acc_u + acc_s} \tag{13}$$

where $acc_s$ is the average per-class top-1 (T1) accuracy of the test images from the seen classes and $acc_u$ is average per-class top-1 (T1) accuracy of the unseen classes. Both of them are computed by Equation (12). For the GZSL setting, we add more recent models for comparison, and the results are presented in Table 4.

**Table 3.** Results of conventional ZSL. The results are reported in %.

| Method | | AWA1 | AWA2 | CUB | SUN |
|---|---|---|---|---|---|
| Embedding-based | DeViSE [15] | 54.2 | 59.7 | 52.0 | 56.5 |
| | DEM [18] | 68.4 | 67.1 | 51.7 | 61.9 |
| | DSEN [19] | — | 72.3 | 71.8 | 62.2 |
| Generation-based | SE-GZSL [26] | 69.5 | 69.5 | 59.6 | **63.4** |
| | ZSM L [27] | **73.5** | **76.1** | 69.6 | 60.2 |
| | f-CLSWGAN [28] | 68.2 | — | 57.3 | 60.8 |
| | Our JG-ZSL | 70.6 | 69.4 | **72.5** | 60.3 |

We compute the harmonic accuracy $H$, corresponding train accuracy $acc_s$, and test accuracy $acc_u$ of our algorithm on all four of the above-mentioned datasets. The results are recorded in Table 4, and all results are cited directly from their published papers.

Table 4 shows that our method achieved high value in both $H - mean$ and $acc_u$ for CUB. Our method shows a significant improvement of 2.9% compared to the second one, and for $acc_u$, we lead the second place by 8.1%. We also achieve the best position for AWA2 on $acc_u$, which leads the second place by 2.5%, and we achieve the second-best position for AWA2 on $H - mean$, while the best Top-1 $H - mean$ is yielded by IZF. For SUN, we achieve the second-best position on $acc_u$ and $H - mean$ and were significantly ahead of the third-best result. These results show that the method presented in this paper has achieved remarkable results.

In addition, it is worth noting that although compared with IZF, the current best SoTA model, our results cannot exceed it in all indicators, the IZF model, as acknowledged by its authors, is based on generative flows and has extremely high complexity, requiring a large number of computational resources and complex computational processes, and it takes human experience and trial and error to obtain the optimal combination of parameters. In contrast, our proposed model is lightweight, simple and easy to train. Similar results can be achieved while consuming far less computing resources than IZF.

### 3.3.2. Ablation Studies

In this paper, we employ a hybrid model combining the generation-based method and embedding-based method and two independent generative networks to synthesize the visual features for each unseen class. While testing, the two generative networks alleviate the problem of data imbalance by synthesizing visual features and semantic embedding, respectively.

In order to illustrate the effects of the multiple generative adversarial networks, we conduct the following experiments on ZSL and GZSL tasks: (1) experiments with only semantic embedding net ($SE$); (2) experiments with semantic embedding net and visual feature generation net ($G_f$); (3) experiments with semantic embedding net, visual feature generation net and semantic embedding generation net ($G_a$); (4) experiments with the whole JG-ZSL. The results are presented in Tables 5 and 6, respectively.

**Table 4.** Results of GZSL on five standard datasets, $U$: T1 per-class accuracy $acc_u$ on unseen class set $U$, and $S$: T1 per-class accuracy $acc_s$ on seen class set $S$, $H$ = harmonic mean. The results report Top-1 accuracy in %, and the best results are marked in bold.

| Method | AWA1 | | | AWA2 | | | CUB | | | SUN | | |
|---|---|---|---|---|---|---|---|---|---|---|---|---|
| | U | S | H | U | S | H | U | S | H | U | S | H |
| BZSL [44] | 19.9 | 23.9 | 21.7 | - | - | - | 18.9 | 25.1 | 20.9 | 17.3 | 17.6 | 17.4 |
| ZSKL[45] | 18.3 | 79.3 | 29.8 | 18.9 | 82.7 | 30.8 | 24.2 | 63.9 | 35.1 | 21 | 31 | 25.1 |
| DEM [18] | 32.8 | 84.7 | 47.3 | 30.5 | 86.4 | 45.1 | 19.6 | 54 | 13.4 | 20.5 | 34.3 | 25.6 |
| CSSD [46] | 34.7 | 87.1 | 49.6 | - | - | - | 19.1 | 62.7 | 29.3 | - | - | - |
| SPF-GZSL [47] | 48.5 | 59.8 | 53.6 | 52.4 | 60.9 | 56.3 | 30.2 | 63.4 | 40.9 | 32.2 | **59.0** | 41.6 |
| TCN [48] | 49.4 | 76.5 | 60.0 | 61.2 | 65.8 | 63.4 | 52.6 | 52.0 | 52.3 | 31.2 | 37.3 | 34.0 |
| SE-GZSL [26] | 56.3 | 67.8 | 61.5 | 58.3 | 68.1 | 62.8 | 41.5 | 53.3 | 46.7 | 40.9 | 30.5 | 34.9 |
| RFF-GZSL [49] | 59.8 | 75.1 | 66.5 | - | - | - | 52.6 | 56.6 | 54.6 | 45.7 | 38.6 | 41.9 |
| IZF [43] | **61.3** | 80.5 | **69.6** | 60.6 | **77.5** | **68.0** | 52.7 | **68.0** | 59.4 | **52.7** | 57.0 | **54.8** |
| NereNet [50] | 56.2 | 70.1 | 62.4 | - | - | - | 51.0 | 56.5 | 53.6 | 45.7 | 38.1 | 41.6 |
| UFG [51] | 59.3 | 66.0 | 62.5 | - | - | - | 45.2 | 56.8 | 50.4 | 35.8 | 46.0 | 40.2 |
| DPR [52] | 54.7 | **81.9** | 65.6 | - | - | - | 48.9 | 66.6 | 56.4 | 8.1 | 35.5 | 40.7 |
| Our JG-ZSL | 57.9 | 63.4 | 60.5 | **63.1** | 68.3 | 65.6 | **60.8** | 63.9 | **62.3** | 50.2 | 37.9 | 43.2 |

**Table 5.** Comparison results with different network options during the testing phase in ZSL. The results are reported in %.

| Method | AWA1 | AWA2 | CUB | SUN |
|---|---|---|---|---|
| $SE - Only$ | 54.3 | 58.5 | 58.4 | 50.1 |
| $SE+G_f$ | 65.9 | 68.1 | 67.6 | 53.7 |
| $SE+G_f+G_a$ | 68.3 | 71.2 | 70.9 | 56.3 |
| $SE+G_f+G_a+L_{SMAC}$ | 70.6 | 69.4 | 72.5 | 60.3 |

From Table 5, it can be seen that the networks have different effects on the datasets, and the JG-ZSL yields the best results on most datasets. For the AWA2 dataset, the accuracy of $T1$ per class on $SE + G_f + G_a$ is higher than the whole JG-ZSL, while the performance is different on the other dataset. Furthermore, from Figure 6, it can be seen that compared with $SE - Only$ and the visual feature generate-only, the JG-ZSL also outperforms all the networks and settings on all the datasets.

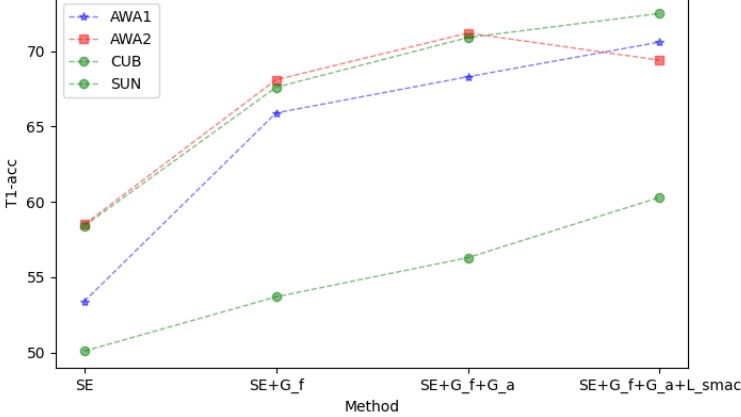

**Figure 6.** Comparison results with different network options on four benchmark datasets in ZSL.

From Table 6, the JG-ZSL yields the best results of harmonic mean on all the datasets. However, because the unseen accuracy $U$ is seriously below the seen accuracies $S$, the accuracy of harmonic mean $H$ is mostly up to unseen accuracy $U$. Therefore, the key to improving the harmonic mean $H$ is to improve the unseen accuracy.

Furthermore, from Figure 7, it can be seen that compared with $SE+G_f$ and $SE+G_f+G_a$, the generated network does not improve the accuracy of the seen class as much as the unseen class, but the JG-ZSL still outperforms all the networks and settings on all the datasets because of the great enhancement to the unseen class.

**Table 6.** Comparison results with different network options during the testing phase in GZSL. The results are reported in %.

| Method | AWA1 | | | AWA2 | | | CUB | | | SUN | | |
|---|---|---|---|---|---|---|---|---|---|---|---|---|
| | U | S | H | U | S | H | U | S | H | U | S | H |
| SE-Only | 21.6 | 55.7 | 31.1 | 21.0 | 59.7 | 31.1 | 36.3 | 44.2 | 39.9 | 19.0 | 27.1 | 22.3 |
| SE+$G_f$ | 50.3 | 60.5 | 54.9 | 50.6 | 62.3 | 55.9 | 52.2 | 59.3 | 55.5 | 35.1 | 23.7 | 28.3 |
| SE+$G_f$+$G_a$ | 54.7 | 61.3 | 57.3 | 54.4 | 69.3 | 61.0 | 57.7 | 63.3 | 60.7 | 51.3 | 36.1 | 42.3 |
| SE+$G_f$+$G_a$+$L_{SMAC}$ | 57.9 | 63.4 | 60.5 | 63.1 | 68.3 | 65.6 | 60.8 | 63.9 | 62.3 | 50.2 | 37.9 | 43.2 |

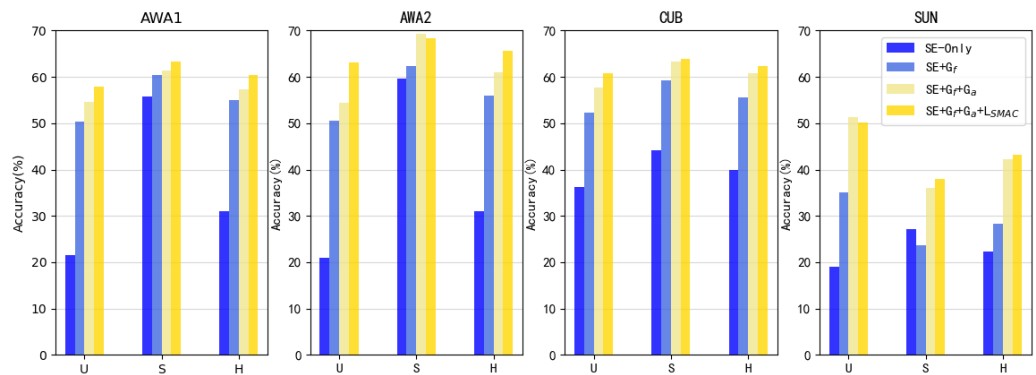

**Figure 7.** Comparison results with different network options on four benchmark datasets in GZSL.

### 3.3.3. Hyperparameter Analysis

We study the balance factor $\gamma$ in Equation (8) to determine its influence on the module, $\gamma$ was set as 0.1, 0.5 and 0.8 in turn, and the ablation results of CUB and AWA2 were shown in Table 7.

**Table 7.** The effectiveness of the balance factor $\gamma$. The results are reported in %.

| $\gamma$ | CUB | | | AWA2 | | |
|---|---|---|---|---|---|---|
| | U | S | H | U | S | H |
| 0.1 | 53.4 | 57.2 | 54.7 | 63.1 | 68.3 | 65.6 |
| 0.5 | 57.9 | 60.1 | 60.0 | 60.8 | 67.7 | 64.1 |
| 0.8 | 60.8 | 63.9 | 62.3 | 60.1 | 66.5 | 63.1 |

As shown in Figure 8, as $\gamma$ grows, $S$, $U$ and $H$ gain consistent improvement on the fine-grained datasets (e.g., CUB), while coarse-grained datasets (e.g., AWA2) do the opposite. This result may reflect that the increase of intra-class compactness can improve the precision of fine-grained datasets, while for coarse-grained datasets, it is necessary to increase the inter-class separability for ambiguous classes.

We then uniformly set the number of generated semantic features to 2 and use generated semantic features and real semantic features to synthesize visual features. Assuming that the total number of synthesized visual features is $N$, the two generated semantic features and real semantic features generate $1/3*N$ synthesized visual features, respectively, and they contrast the effects by varying the number of visual features generated. The ablation results of CUB and AWA2 were as follows:

The number of generated samples is an important part for generative methods, so we also implement detailed experiments in this area. Our method achieves the best results on AWA1, AWA2, CUB, and SUN when we synthesize 1800, 2400, 600 and 90 examples per unseen classes, respectively. Figure 9 shows part of the experimental results, and there is an obvious phenomenon here that the number of generated samples for unseen classes is positively correlated with the values of U and H, which shows that the data-imbalance problem has been relieved by the generation model in our framework. However, with the large increase of unseen generated samples, the classification accuracy of seen classes also decreased significantly, which is one of the future directions to explore.

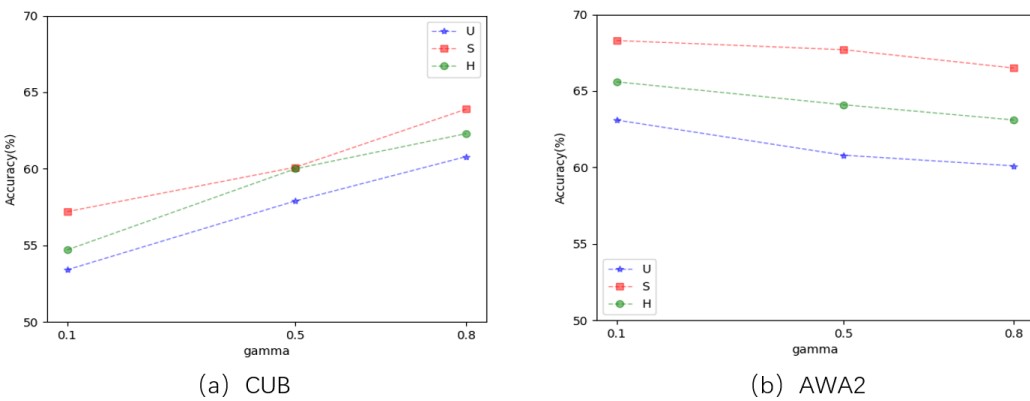

| (a) CUB | (b) AWA2 |

**Figure 8.** The effectiveness of the balance factor $\gamma$.

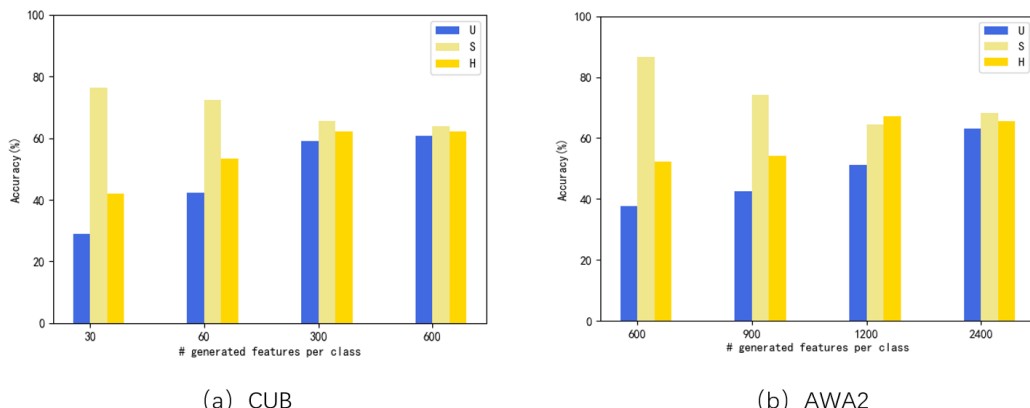

| (a) CUB | (b) AWA2 |

**Figure 9.** The influence of different numbers of the synthesized samples for each unseen class.

In this paper, the generalization of the $G_f$ and the specificity of the $G_a$ are combined not only to improve ZSL performance but also to alleviate the data imbalance problem and reduce the gap between seen accuracy $S$ and unseen accuracy $U$ and improve GZSL performance.

## 4. Conclusions

In this paper, we propose a joint model with multiple generative adversarial networks combining the embedding-based method and the generation-based method to synthesize the visual features and the semantic embedding points which realized the data enhancement of zero-shot learning in two ways, and it is also verified in the more challenging generalized zero-order learning setting. Inspired by the ideas of active learning and generative adversarial networks, the coupled generative networks work cooperatively to synthesize visual features of unseen classes under the constraint of semantic-relevant self-adaptive margin center loss. In addition, we compare the model with the current advanced

methods, and the experimental results outperform the state-of-the-art embedding-based method and are competitive with the current generation-based method.

However, there are still some limitations in this paper. For example, all categories use the same way to generate semantic features which are not targeted enough, and there is no attempt to use VAE and other models to generate semantic features for comparison. Making full use of the pseudo-semantic features generated by images and comparing them with more generation models is the direction of future exploration. In addition to the above problem, exploring the more appropriate number of generated semantic features and different proportions of generated samples synthesized by generated semantic features and real semantic features are also problems that can be explored. In future work, we will further explore more efficient pseudo-semantic features generation methods and explore more obvious ways to improve the effect for unseen class classification and conduct experiments on larger datasets to improve the generalization abilities.

**Author Contributions:** Conceptualization, M.Z. and Y.S.; methodology, M.Z.; investigation and validation, M.Z., X.W. and Y.S.; data curation and formal analysis, M.Z., X.W. and W.W.; writing—original draft preparation, M.Z. and Y.S.; writing—review and editing, S.R., X.W. and W.W.; funding acquisition, S.R. and W.W. All authors have read and agreed to the published version of the manuscript.

**Funding:** This research received no external funding.

**Data Availability Statement:** The dataset used in this paper is publicly available at [42].

**Conflicts of Interest:** The authors declare no conflict of interest.

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
