# Peer review of "Zero-Shot Learning with Joint Generative Adversarial Networks"

_electronics, doi:10.3390/electronics12102308_

Round 1

Reviewer 1 Report

The manuscript by Zhang et al entitled "Zero-Shot Learning with Joint Generative Adversarial Networks" represents the new approach comprising at least two important directions in machine learning: zero/one-shot learning and generative modelling.  The manuscript is clearly written in general, however, several recommendations have to be made.

1.  "Finally, semantic-relevant self-adaptive margin center loss was used, which can explicitly encourage intra-class compactness and inter-class separability, and guide coupled GAN to generate discriminative and representative semantic features." The term "metric learning" is closely related to this formulation, which is, however, is not mentioned in the manuscript despite its visibility in few-shot learning in general. Could you, please, provide the extra explanations on this.

2. It seems that some key references were missed:

Lake et al One Shot Learning of Simple Visual Concept. CogSci (2011)

Wang et al Generalizing  from a Few Examples: a Survey on Few-Shot Learning ACM Computing Surveys (2020) 53, 3, 63

Vinyals et al  Matching Networks for One Shot Learning. In advances in Neural Information Processing Systems (2016) 1473

3. In Abstract the Authors have mentioned the data imbalance problem, which is, however, is not on the march in the main body of the manuscript. I would suggest you to briefly discuss it or even to introduce the specially adopted measures more extensively that it has found its place currently with the corresponding  explanations made.

4. It may be assumed that Transfer Learning should be enumerated in the Introduction among others.

5. "Zero-sample" is it a typo? (zero-shot)

6. "...which is obviously something that machines don't have." Please, re-formulate the sentence to avoid these "judgements" as the excessive ones in this case.

7. "Although the traditional mapping method is adopted in this paper, a non-linear projection is added to obtain a better embedding effect."

The sentence should be clarified.

8. "The commonly-used semantic embedding methods rely on a structured loss function proposed in [29]"

Please, verify this ref. My search for the structured loss function in this work was unsuccessful.

9. Eq.(1): Does this equation describe the metric learning procedure? Additionally, the embedding space  in (1) is linear, non-linear or kernel-defined?

10.  "Chen et al [30] introduced a learnable non-linear projection that improves the quality of the learned semantic embedding" was straddled with "Instead of using the original method, we limit our
embedding space to the semantic descriptor space, so the dimension of this space is equal to the dimension of the semantic descriptor."

The origin of this improvement may be of complex character in this case. Is it possible to verify this?

11.  The class imbalance was not described (the text above eq.(2))

12. Eq. 2: The temperature parameter should be described. Is it kNN-related or learning rate-related or something very directly defining?

13. "However, the methods above generally assume that the sample is subject to some distribution (generally Gaussian distribution), which also leads to a large deviation between the generated sample and the real sample, and cannot truly represent the real data situation of the unseen class".

Is it correct that authors suppose to generate the samples from other distributions or they assume the distribution as non-Gaussian?

14. "Gans"

15. The discussion of the following studies on generative modelling may be in place:

Li et al Adversarial Feature Hallucination Networks for Few-Shot learning. CVPR (2020) 13470.

Bond-Taylor et al Deep Generative Modelling: a Comparative  Review of VAEs, GANs, Normalizing Flows, Energy-Based and Autoregressive  Models. IEEE Transactions of Pattern Analysis and Machine Intelligence (2022) 44, 11, 7327

15. Ref. [32] corresponds to the improved variant of WGAN not the original one. Please, precise which version you have used. Besides, the problem of the stabilizing training is discussed in [Bond-Taylor et al] (please, see above), this review may be involved as a good support  for the paragraph related to Methodology.

16. "Loss design" Is it a typo?

17. In the paragraph Classification, some parameters are not defined.

18. Paragraph "Conclusions" has to be extended. Please, pay your attention that "active learning" is mentioned only here.

Reviewer 2 Report

This paper have introduced an innovative working mechanism (the SBC memory) and surrounding infrastructure (BitBrain) based upon a novel synthesis of ideas from sparse coding, computational neuroscience and information theory that support single-pass learning, accurate and robust inference, and the potential for continuous adaptive learning. The authors have demonstrated the efficacy of these concepts on the MNIST and EMNIST benchmarks and shown that the proposed inference mechanism has very low training costs and is robust to noise.  The paper is organized well. However, there are some points to be considered.

1.  Abstract needs to be more precise highlighting major contributions.

2.  It would be nice to explicitly list the future research directions. Please also discuss some limitations of your proposed method.

3. The background of the proposed study should be further explained in detail. Some concepts are hard to comprehend without explaining clearly.

4. Grammar is expected to be further improved. Please check the manuscript carefully to remove the typos, improve the language and format.

5. Please test the robustness of the proposed method.

Reviewer 3 Report

Dear authors, 

I found your manuscript a valuable approach to classify unseen classes.

Although its performance is fair, I consider it could be improved a bit, I got a feeling that you did not explore until the end, but this is just my impression, I am not judging your effort, because you compare it against several datasets, with huge attributes, but maybe the ablation studies help to explore deeper.

Also, the performances are variable, some times is the best, others second, and so on, which maybe could be an indicator that something is missing.

I would suggest to describe or illustrate some images from each dataset, showing their differences or similarities.

It is all for this time. 

Best regards, 

Reviewer 4 Report

This paper presents a method for zero-shot image classification. The authors propose the usage of two conditional GANs. The first GAN receives as input a class attribute and generates visual features. Generated visual features of unseen classes form an average class visual feature. The second GAN takes as input this average class visual feature and generates semantic embeddings of the class. Then, this embedding is the input of the first GAN to generate synthetic visual features. Those features of the unseen classes and the ones from the seen classes are mapped to a common embedding space and they are employed to train a classifier for image classification. Moreover, different losses are utilised: a semantic embedding loss to learn visual to semantic mapping, a Wasserstein loss for GAN training and self-adaptive margin centre loss to achieve intra-class compactness and inter-class separability. 

The idea is, to my knowledge, novel. The introduction is well structured and it makes a good presentation of the problem. However, some recent works should be added. Only one work of 2021 exists and no work of 2022. The proposed method of the paper is, generally, well presented. Some points that, in my opinion, should be clarified:

  • The classification section (section 2.4) could be accompanied by a flow diagram or an algorithm. The workflow would be clearer, text only presentation may be a bit confusing.

  • In the same section it is not clear to me the difference between x̃ and the output of G1 (at the definition of the Utr).

  • The meaning of the term “semantic embedding interval” is not clear, it should be further explained.

  • The term ei in Figure 1 is not defined. 

  • The explanation of the constraints of non-linear space to semantic embedding space I think is rather confusing, both in Figure 1 and at the last paragraph before section 2.2.2.

Finally, the experimental results are the weak point of this paper. No comparison is made with methods after 2020. The results presented are not satisfying. The proposed method is SoTA only in the CUB dataset. In the other three datasets, big differences exist in the results (3 to 11 %). Moreover, Table 5 and Figures 4 and 5 demonstrate the same results in different ways. An ablation study could also be made for some parameters of the losses (e.g., γ and te) for a more enriched experimental section.

Reviewer 5 Report

The paper is fine and worthy of publication.  On page 3 the statement "out approach has better classification performance than other methods" is clearly not quite accurate and should be removed or changed to say is competitive with other methods.  On page 1, "fast-developing developing field" should be "fast-developing field".  Pedantically "et al" should be "et al."  On page 5 "Gans" should be "GANs"

Round 2

Reviewer 2 Report

I am satisfied with the revision. No comments remained.

Author Response

Thanks for the positive comments.

Reviewer 3 Report

Dear authors, 

After check my suggestions were taken into account, I consider it enough, to approve its content.

I would recommend some minor change on figures 6 and 7, I mean, they should use the same scale or range in axis, because it seems you try to overqualify the performance. I would prefer to use 0-100% or at least the same range when you compare two figures side by side.

That is all, I wish you the best.

Reviewer 4 Report

The authors addressed the majority of my comments. However, the main drawback of the paper still exists: The proposed method doesn't have SoTA results (or close to the SoTA) in most datasets. Thus, I think this work is not ready for publication.

Round 3

Reviewer 4 Report

The authors are right that their method is simpler and less computationally intensive than the IZF model. Nevertheless, this is a rather old method. The proposed method has also inferior results in more recent methods (e.g., [1]), in both unseen and seen classes.

Moreover, the results are far from the SoTA in the ZSL task also (Table 3).

I do not think that this work has publishable results.

[1] Kong, X., Gao, Z., Li, X., Hong, M., Liu, J., Wang, C., ... & Qu, Y. (2022). En-compactness: Self-distillation embedding & contrastive generation for generalized zero-shot learning. In Proceedings of the IEEE/CVF Conference on Computer Vision and Pattern Recognition (pp. 9306-9315).